# Targeting Immune Modulators in Glioma While Avoiding Autoimmune Conditions

**DOI:** 10.3390/cancers13143524

**Published:** 2021-07-14

**Authors:** Lynn Bitar, Ulrike Schumann, Renate König, Frauke Zipp, Mirko H. H. Schmidt

**Affiliations:** 1Department of Neurology, Focus Program Translational Neuroscience (FTN) and Immunotherapy (FZI), Rhine Main Neuroscience Network, University Medical Center of The Johannes Gutenberg University Mainz, 55131 Mainz, Germany; lynn.bitar@unimedizin-mainz.de; 2Institute of Anatomy, Medical Faculty Carl Gustav Carus, Technische Universität Dresden School of Medicine, 01307 Dresden, Germany; ulrike.schumann@tu-dresden.de; 3Host-Pathogen Interactions, Paul-Ehrlich-Institute, 63225 Langen, Germany; Renate.Koenig@pei.de

**Keywords:** autoimmune disease, glioma, immune checkpoints, immunotherapy, clinical trials

## Abstract

**Simple Summary:**

Glioblastoma multiforme is a futile disease usually leading to the patient’s death within one year post-diagnosis; therefore, novel treatment options are desperately needed. In this regard, activation of the inert immune system has moved into focus in recent years. Malignant brain tumors, as well as autoimmune diseases, elicit aberrant immune responses. In this way, glioma escapes the host’s immune system and, thus, activation of the immune response in order to reduce tumor tolerance can serve as an alternative treatment option. Immune checkpoint modulators in combination with traditional therapies have gained attention in both glioma and autoimmune diseases. In this review, we highlight ongoing or completed clinical trials that target immune modulators in these diseases.

**Abstract:**

Communication signals and signaling pathways are often studied in different physiological systems. However, it has become abundantly clear that the immune system is not self-regulated, but functions in close association with the nervous system. The neural–immune interface is complex; its balance determines cancer progression, as well as autoimmune disorders. Immunotherapy remains a promising approach in the context of glioblastoma multiforme (GBM). The primary obstacle to finding effective therapies is the potent immunosuppression induced by GBM. Anti-inflammatory cytokines, induction of regulatory T cells, and the expression of immune checkpoint molecules are the key mediators for immunosuppression in the tumor microenvironment. Immune checkpoint molecules are ligand–receptor pairs that exert inhibitory or stimulatory effects on immune responses. In the past decade, they have been extensively studied in preclinical and clinical trials in diseases such as cancer or autoimmune diseases in which the immune system has failed to maintain homeostasis. In this review, we will discuss promising immune-modulatory targets that are in the focus of current clinical research in glioblastoma, but are also in the precarious position of potentially becoming starting points for the development of autoimmune diseases like multiple sclerosis.

## 1. Introduction

Today, we know there is a short- and long-range extensive communication between the nervous and the immune system [1]. The immune system shapes processes of the nervous system and, in return, the nervous system regulates immune function in the rest of the body. If this balance is disrupted, neuroinflammation occurs. Variable cues can initiate such responses, including primary malignant brain tumors (glioma), infections, ischemic stroke, toxic metabolites, or traumatic brain injury. Ultimately, neuroinflammation can lead to autoimmunity and neurodegenerative diseases [2,3,4,5]. In autoimmune disease research, but also central nervous system (CNS) oncology, this imbalance in immune response is associated with multiple known immune checkpoint molecules (Figure 1). Autoimmune diseases are typically characterized by the presence of autoreactive immune cells and the production of autoantibodies. On the other hand, the most common malignant CNS tumor, glioblastoma multiforme (GBM), is characterized by its immune evasion mechanisms that result in a poor prognosis and death, usually within one year postdiagnosis [6,7]. Immunomodulatory therapies, targeting a narrow set of immune pathways, have been widely implemented in cancer treatment in the past decade [8,9,10]. A challenge in cancer immunotherapies is to contain the collateral appearance of autoimmunity. In turn, glioma progression may loop back, adding to the intensity of the underlying inflammation [11]. Therefore, the risk of brain tumor development in relation to pre-existing autoimmune diseases should be kept in mind. While some studies observed a reduced risk of glioma in patients with autoimmune disorders [12], others did not [13,14]. However, subgroup analyses of patients who were younger than 40 years old revealed a positive association between pre-existing inflammatory bowel disease and risk of glioma, and a negative association between asthma and glioma incidence [13]. The reduced incidence of gliomas in patients with autoimmune disorders might be due to the activated immune response against cells and, thereby, glioma cells. On the other hand, patients with concomitant autoimmune diseases are often excluded from clinical trials involving immune checkpoint inhibitors, as the treatment could lead to the development of severe life-threatening events, such as exacerbation of the underlying immune condition [15]. The risk–benefit ratio of an exclusion of these patients needs to be evaluated, as adverse effects may be managed with corticosteroids or other immunosuppressive therapies. An immune checkpoint inhibitor treatment for patients with pre-existing autoimmune disorders could be safe if carefully monitored [16].

Here, we first provide a brief overview of the relevant signaling molecules and immune modulators, and then discuss the clinical trials that target these, with a specific focus on glioblastoma and autoimmune diseases.

## 2. Immune Checkpoints Overview

Currently used immunotherapies eliminate tumor cells by enhancing the body’s autoimmune function. They consist of (1) checkpoint immunotherapies, (2) active immunotherapies using cancer vaccines and immune stimulatory gene therapy, (3) passive immunotherapies using antibodies, and (4) adoptive immunotherapies using chimeric antigen receptor (CAR) T cells [17]. A strong immune response to such therapies holds the potential for autoimmune events, including autoimmunity directed at the CNS [18]. Immune cells rely on one or more cell surface signaling molecules to initiate an immune response. After the primary surface receptor signal starts, the secondary signal, co-signal, modulates the immune response by inhibiting or stimulating cell communication. In T lymphocytes, the primary signal is mediated by a specific T cell receptor (TCR), which binds to a major histocompatibility complex (MHC)–peptide structure on a professional antigen-presenting cell (APC). The activation does not occur solely through TCR-mediated signaling; without a second signal, T cells fall into a nonresponsive state (anergy) in which they fail to respond [19]. T cells are the most comprehensively and extensively studied cell type regarding the role of co-signals.

Surface receptors, such as immune checkpoint molecules, and cytokines, such as transforming growth factor beta (TGF-β) or interleukin-6 (IL-6), are important for differentiation of T cells, especially CD4-positive cells [20]. In the past several decades, various inhibitory immune checkpoint molecules have been identified and studied in cancer, including programmed death-1 (PD-1), cytotoxic T-lymphocyte antigen 4 (CTLA-4), lymphocyte activation gene-3 (LAG-3), T-cell immunoglobulin and mucin-domain-containing protein 3 (TIM-3), and T cell immunoreceptor with Ig and ITIM domains (TIGIT). Extensive attempts to target these immune checkpoint molecules have rendered them ‘classical’ targets.

Furthermore, T helper (Th) cells have a central role in modulating immune responses. While Th1 and Th2 cells have long been known to regulate cellular and humoral immunity, Th17 cells have been identified more recently as a subset of proinflammatory Th cells, defined by their production of IL-17. They play an important role in glioma progression and have also been implicated in the pathogenesis of many inflammatory and autoimmune diseases [21,22]. IL-17A supports proliferation of cancer cells by stimulating fibroblasts to upregulate the vascular endothelial growth factor (VEGF), resulting in tumor neovascularization [23,24]. VEGF is overexpressed in most solid tumors and is a popular target for antiangiogenic agents. However, VEGF suppression is not effective in all cancers, and often shows limited ability to ameliorate the overall survival in patients [25]. There also exists an inter-relationship between the VEGF system and various autoimmune diseases, such as rheumatoid arthritis (RA) and multiple sclerosis (MS), making it a valuable target [26].

Another subset of Th cells, regulatory T cells (Tregs), are functional antagonists to Th17 cells due to their suppressive effector function through secretion of inhibitory cytokines, such as IL-10 and TGF-β, or through cell-mediated engagement of inhibitory checkpoint molecules, such as TIGIT and CTLA-4. A lack of Tregs can result in lethal autoimmunity, whereas an increase in Tregs is often associated with tumor progression and reduced survival in cancer patients. Moreover, the balance between Th17 and Tregs is critical for maintaining homeostasis, which is tightly regulated via the TGF-β/IL-2 and IL-6 cytokine axis [27]. A major regulator of the Th17–Treg axis is the highly conserved serine/threonine kinase casein kinase II (CK2). This nonclassical immune modulator has been historically studied in the context of cancer, but is also relevant for many T-cell-driven autoimmune disorders, including MS [28]. Among other immune cells, T cells express tumor necrosis-factor-related apoptosis-inducing ligand (TRAIL) at the surface, which can induce apoptosis by binding to its cognate receptors [29]. As a major cytokine of the tumor necrosis factor (TNF) superfamily, it is a protein of interest in treating glioma and autoimmune diseases.

Natural killer (NK) cells often utilize multiple activating receptors to transmit a primary signal. The rapid activation of NK cells is controlled by a large number of different coinhibitory molecules. Since NK cells share a common progenitor with T cells, and in many aspects are very closely related to T cells, many surface molecules are shared between NK cells and T cells [20]. For instance, LAG-3, TIM-3, TIGIT, and TRAIL are also expressed on NK cells.

B cell activation requires, apart from a primary signal mediated by the B cell receptor (BCR), a secondary signal mediated by cytokines or surface receptors, such as CD40 or CD27, through interaction with T cells [20]. The surface receptor CD20 was reported to be physically and functionally coupled to MHCII and CD40, which are both critical for B and T cell interactions. CD20 participates in BCR signaling, either by acting as a calcium channel and being involved in B cell activation, or by directly modulating the BCR [30]. Additionally, PD-1, LAG-3, and TIM-3 are also known B cell coinhibitory surface receptors.

Myeloid-derived phagocytes, such as macrophages, monocytes, and dendritic cells, as well as mast cells, utilize co-surface receptors for activation and modulation of the immune response. For instance, PD-1 and TIM-3 can be found on macrophages and dendritic cells [20].

Immune checkpoint molecules are defined as ligand-receptor pairs that employ inhibitory or stimulatory effects on immune responses [31]. Immune checkpoint inhibitors evolved in humans as part of regulatory circuits to quickly halt an immune response when reacting to foreign antigens, so that the immune system does not harm the body itself. Several tumors hijack these regulatory circuits to prevent an effective antitumor response, but, if the immune checkpoint inhibitors are knocked out in animal models or blocked therapeutically in patients, autoimmune diseases can develop [32]. The major therapeutic target in autoimmune diseases is an immune intervention targeting costimulatory pathways in immune cells [33]. Retrospective data largely suggest that patients with autoimmune disease may benefit from immunotherapy.

Tumors are known for modulating immune checkpoint molecules to avoid immune detection. The tumor microenvironment is tightly involved in the local immune response due to this modulation [34]. In particular, this crosstalk occurs in the extracellular space among tumor and immune cells, e.g., microglia, macrophages, or lymphocytes, but also with stromal cells, such as fibroblasts, endothelial cells, or even the noncellular components of the extracellular matrix (ECM). The capability of tumor immune evasion is dictated by the interaction with intrinsic and extrinsic secreted components of the tumor, along with the cytokines and chemokines of the tumor microenvironment [35]. To date, the success of immunotherapies in animal models has not always been replicated in clinical trials and has been met with translational limitations [36,37]. Here, we summarize clinical trials targeting classical (Figure 2) and nonclassical immune modulators in glioblastoma and autoimmune diseases whose outcome relies on immune responses.

## 3. Classical Immune Checkpoint Molecules: Efficiency and Limitations

### 3.1. PD-1 (CD279)

PD-1 is an inhibitory receptor that belongs to the CD28 family. The receptor is expressed mainly on activated T cells [38]. PD-L1, the ligand of PD-1, is expressed on B lymphocytes and APCs, as well as on different types of tumor cells [39]. An engagement of PD-1 with its ligands, mainly PD-L1, causes an inhibition of T cell proliferation, activation, cytokine production, altered metabolism, and cytotoxic T lymphocytes (CTLs) killer functions. Eventually, this causes apoptosis of activated T cells. The T cell response has to be controlled to limit tissue damage and maintain self-tolerance [31].

In glioma, tumor cells hijack the inhibitory pathways controlling T cell response via the PD-1/PD-L1 axis by overexpression of PD-L1. To treat heterogeneous glial tumors, the blockage of immune checkpoint molecules like PD-1 has been challenging, but at least one problem was identified and tackled; namely, that the timing of administration of these inhibitors impaired their efficacy [34]. Recently, in two clinical trials, the neoadjuvant PD-1 blocker nivolumab was pre- and postoperatively administered to patients suffering from primary and recurrent glioma (NCT02550249). In these small cohort studies, the treatment improved local immunomodulatory effects by lifting the suppressive signal on immune infiltrates, yielding improved overall survival and progression-free survival of the patients [40,41]. Additional larger scale prospective studies are needed to evaluate the high value of these trials.

Due to the heterogeneity of glial tumors, numerous clinical trials combine PD-1 inhibitors with classical antitumor therapies, such as chemo- and/or radiotherapy. For instance, in the phase II trial NCT04195139, newly diagnosed elderly glioma patients received a combination of the PD1 antibody nivolumab and chemotherapeutic temozolomide (TMZ), or TMZ alone. A phase III trial (NCT02667587) in glioblastoma patients that present with a methylation of the O6-methylguanine-DNA methyl transferase (MGMT) gene promotor received a combination of nivolumab, TMZ, and radiotherapy. MGMT gene promotor methylation has been investigated as a potential biomarker due to its sensitivity to TMZ treatment. TMZ given concomitantly with radiotherapy, followed by sequentially as single agent, showed superiority over radiotherapy alone [42]. Both trials are ongoing and aim to prove whether nivolumab in combination with other therapies improves the overall survival of GBM patients. Interestingly, resistance to therapeutic blockade of the extensively studied checkpoint inhibitor PD-1 was associated with an upregulation of alternative immune checkpoint molecules, such as TIM-3 [43]. PD-1 is a promising target for supporting first-line therapy, but usage of monoclonal antibodies (mAbs), such as nivolumab, has to be closely monitored for efficacy and side effects. It is also noteworthy that recent findings highlighted a role of PD-1 in immune tolerance as the loss of PD-1-induced autoimmune diseases, such as the CNS-targeting disease MS [39,44]. A study of high dose immune reconstitution in MS patients after autologous hematopoietic stem cell transplant revealed that an early expansion of PD-1-expressing CD8-positive T cells and PD-1-expressing CD19-positive B cells was associated with favorable neurological outcomes, restoring immune tolerance in MS patients caused by PD-1-inhibitory signaling [45]. Additionally, in the *PDCD1* gene, single nucleotide polymorphisms have been reported in patients suffering from peripheral autoimmune disorders, such as RA [46], type 1 diabetes (T1D) [47], and systemic lupus erythematosus (SLE) [48].

### 3.2. CTLA-4 (CD152)

CTLA-4 is a structural and functional homolog of the costimulatory receptor CD28, but acts as a negative regulator of T cell activation. It binds the B7 family molecules CD80 and CD86 on APCs. It is constitutively expressed in Tregs, but only upregulated in conventional T cells after activation, and plays a critical role in the maintenance of tolerance to self-antigens [49].

Blockage of CTLA-4, either alone or in combinatorial treatments, has proven to be highly successful in tumors like melanoma and renal cell carcinoma [38,39,40]. The expression of CTLA-4 in glioma specimens of patients who underwent neurosurgical resection indicated that higher CTLA-4 expression in the tumor microenvironment resulted in greater immune cell infiltration and correlated with a shorter overall survival [41]. Thus, CTLA-4 is a promising novel target for glioma treatment. Recruitment of glioma patients for phase I, II, and III trials using the CTLA-4 inhibitor ipilimumab (a mAb) in combination with a PD-1 inhibitor is ongoing (NCT04323046, NCT04396860, NCT04003649, NCT03233152, NCT04145115). Additionally, targeting recently identified associated molecules and checkpoint receptors may enhance the efficacy of CTLA-4 inhibitors. TIGIT and CD96 are coinhibitory receptors that, together with the costimulatory receptor CD226, form a pathway that is analogous to the CD28/CTLA-4 pathway [50,51]. However, elimination of CTLA-4 may result in the breakdown of immune tolerance and the development of autoimmune diseases [52]. Genetic association studies identified polymorphisms in the CTLA-4 gene that are linked to MS susceptibility [53]. Abatacept, a CTLA-4–Ig fusion protein that blocks the CD28-mediated costimulatory signal necessary for T-cell activation, has been tested in phase I clinical trials for several autoimmune diseases. The administration was well tolerated by patients and revealed an improved overall disease outcome that correlated with decreased T-cell infiltrates in patients suffering from MS, RA, or psoriasis [54,55,56,57,58]. Different mechanisms were proposed for the action of CTLA-4–Ig, including a shift of the immune response toward Th2 in Th1-mediated diseases or the regulation of the tryptophan catabolism in dendritic cells (DC), causing an inhibition of T-cell proliferation [58].

### 3.3. LAG-3 (CD223)

The inhibitory coreceptor LAG-3 is a transmembrane protein with structural similarities to CD4 that is expressed on activated T cells, natural killer T (NKT) cells, NK cells, and B cells [59,60,61,62]. Persistent antigen stimulation in cancer or chronic infection leads to chronic LAG-3 expression, promoting T cell exhaustion. Depleting LAG-3 is possible by application of the anti-LAG-3 mAb GSK2831781 or by the agonistic antibody IMP761 [63].

LAG-3 is expressed in gliomas with a particularly active immune microenvironment [64]. Two separate phase I clinical trials in glioma patients are ongoing, where a combination of LAG-3-specific blocking mAbs with PD-1 inhibitors has been used (NCT02658981, NCT03493932). The co-expression of LAG-3 with PD-1 on tumor-infiltrating lymphocytes (TILs) has led to extensive research on the synergistic blockade of both receptors to trigger an antitumor immune response [65]. Currently, clinical trials are in preparation to assess the beneficial effects of anti-LAG-3 mAbs in autoimmune diseases, including MS (patent no. 3344654) [66]. There are various therapeutic regimens conceivable for this target, which might improve clinical outcome in glioma without shifting the balance to autoimmune disease.

### 3.4. TIM-3 (CD366)

TIM-3, another regulatory immune checkpoint molecule, can be expressed by multiple immune cell types, including CD4-positive Th1 cells, Tregs, B cells, mast cells, NK cells, and myeloid cells, such as DCs and macrophages [67,68,69,70,71].

As TIM-3 was found to be highly expressed in glioma cells isolated from GBM patients, it became a promising target for glioma patients who are resistant to classical immunotherapies [72]. TIM-3 and PD-1 have been shown to be overexpressed on TILs, which exhibit an exhausted phenotype, as defined by failure to proliferate and produce IL-2, tumor necrosis factor alpha (TNF-α), and interferon gamma (IFN-γ) [73]. The overlap in expression and function suggests that both immune checkpoint molecules co-operate to stimulate effector cell exhaustion and thereby indirectly promote tumorigenesis. Kim et al. [74] showed that the blockade of both immune checkpoint receptors, combined with radiation, resulted in a significant increase in survival using a murine glioma model. An ongoing first phase I study in patients with recurrent glioma (NCT03961971) is evaluating the use of the TIM-3 inhibitor MBG453 in combination with anti-PD-1 treatment and radiosurgery. The efficacy of TIM-3 therapy in the treatment of other cancers, like acute myeloid leukemia, validates its potential as a therapeutic target in glioma [75]. Due to the high pathogenicity of CD4-producing IL-17 T cells in autoimmune diseases, a considerable effort has been made to elucidate their regulatory molecules and pathways. Both Th1 and Th17 cells express TIM-3, but Th17 at a lower level [76,77]. CD4-positive T-cells isolated from the cerebrospinal fluid of MS patients displayed an inverse correlation between the expression of IFN-γ and TIM-3, indicating that TIM-3 is dysregulated in MS [78]. This supports the therapeutic value of TIM-3 for the treatment of glioma, as well as autoimmune diseases.

### 3.5. TIGIT (Vstm3, WUCAM)

Comparable to LAG-3 and TIM-3, TIGIT is a checkpoint inhibitory molecule that is expressed on a variety of immune cells, including CD4- and CD8-positive T cells and NK cells [79,80,81,82]. The three ligands of TIGIT, namely CD155, CD112, and CD113, all belong to the family of nectins and nectin-like molecules, and are involved in cell adhesion, cell polarization, and tissue organization [83]. TIGIT binds CD112 and CD113 with lower affinity than CD155 [79,81,82].

CD155 and CD112 are not only expressed on DCs and macrophages, but also highly expressed in various cancer cell lines and tumor specimens [80,84,85,86]. Previous studies revealed an elevated expression of CD155 on human glioma cells and an increased TIGIT expression in patient-derived CD8-positive TILs, which offers this signaling pathway as a potential therapeutic target [87,88]. Blocking TIGIT and PD-1 in a murine glioma model resulted in an increase in IFN-γ-expressing CD8-positive T cells and a decrease in Tregs in the brain, as well as an improved overall survival [89]. In early phase I clinical trials in patients with recurrent glioma (NCT04656535) and in patients with advanced solid tumors (NCT03628677), the safety and efficacy of co-blocking TIGIT and PD-1 using the mAbs AB154 and AB122 are being evaluated. Contrary to the high expression of TIGIT in TILs isolated from glioma patients, T cells isolated from MS lesions showed no expression of TIGIT [90]. The TIGIT/CD226 pathway has been linked genetically to several autoimmune diseases, including MS, RA, and T1D [91]. CD226 competes with TIGIT for binding to the same ligands but delivers a positive stimulatory signal to immune cells. In addition to the regulation of DCs, TIGIT suppresses the T cell response in a direct T cell intrinsic manner. TIGIT knock-out mice are more susceptible to the development of spontaneous experimental autoimmune encephalomyelitis (EAE), suggesting an important role for the CD226/TIGIT pathway in autoimmune responses [92]. Furthermore, isolated T cells from MS and T1D patients showed inhibited activation and proliferation when treated with neutralizing anti-CD226 mAbs ex vivo [93]. Therapeutic approaches targeting CD226 in autoimmune diseases exclusively affect proinflammatory Th1 and Th17 cells because naïve T cells do not express CD226. The opposing pattern of TIGIT expression in glioma and MS patients hints that anti-TIGIT therapy may indeed be beneficial for patients with GBM [90].

## 4. Pathogenic Infiltrating Th17 Cells

The above-mentioned receptors are the most potent examples of T cell immune checkpoint molecules. These evolutionarily conserved negative regulators of T cell activation are involved in the fine-tuning of immune response and activity [94]. Indeed, therapies relying on the chimeric antigen receptor using, for instance, T cells are under investigation in glioma and autoimmune disease [95,96]. The T cell subset Th17 has emerged as a key player in host defense contributing to glioma progression and the pathogenicity of autoimmune diseases [21,22]. Owing to their plasticity, Th17 cells may switch to become ex-Th17 cells (or nonclassical Th1 cells) that no longer produce IL-17, but rather produce IFN-γ. These cells are characterized by an increased production of proinflammatory cytokines and are mostly resistant to the suppression of proliferation and cytokine production mediated by Tregs [97,98]. Ex-Th17 cells have been shown to accumulate and play a role in multiple autoimmune disease models, including those for RA or MS [97,99,100]. Depending on cytokine availability in the tumor microenvironment, a pleiotropic role of Th17 cells in tumor progression has been proposed. Th17 cells may be tumor-cytolytic, driving an antitumor immune response by the expression of high levels of IFN-γ [101]. On the other hand, Th17 cells may also elicit a protumor immune response by exerting immunosuppression via TGF-β1-induced IL-10 secretion [102,103]. Th17 cells modulate glioma growth depending on the cytokines produced locally [104].

Another population of cytotoxic CD8-positive T cells producing IL-17, termed Tc17 cells, has been shown to generate Th17 cells and render them more encephalitogenic in the MS model [105]. In fact, Tc17/IFN-γ cells are commonly detected in inflamed human or mouse tissues, as well as in peripheral blood in MS, psoriasis, SLE, and RA, proposing an involvement in autoimmune diseases [106,107,108,109]. Furthermore, a potent antitumor efficacy of Tc17 cells has been reported in some tumors [110,111].

## 5. Nonclassical Immune Modulators

### 5.1. CK2 

CK2 is a protein kinase governing cell cycle progression and survival [28]. Due to its function as an intrinsic regulator of CD4-positive effector T cells and the regulation of the Th17/Treg balance, it is widely recognized as an established immune modulator [112,113]. Moreover, CK2 controls the Th2 inflammatory responses by Tregs [114].

In glioma patients, CK2 is expressed in all tumor cells, where it supports cell survival [115]. An ongoing trial applies the ATP-competitive specific CK2 inhibitor CX-4945 (silmitasertib sodium) to children with recurrent medulloblastoma (NCT03904862). Additionally, this particular CK2 inhibitor caused cell death in multiple myeloma and lymphoma cells isolated from patients [116]. Finally, malignant solid tumors, where aberrant epidermal growth factor receptor (EGFR)-mediated and nuclear factor kappa-light-chain-enhancer of activated B-cells (NF-κB) signal transduction pathways are responsible for resistance to conventional therapies, are known to be regulated by CK2 [117,118]. A delayed NF-κB activation has been reported to contribute to the therapeutic resistance of some malignant gliomas to CK2 inhibition [115]. Interestingly, CK2 may regulate the expression of EGFR itself, as shown by its downregulation in response to CK2 inhibition [118]. As CK2 has been associated with antitumor drug resistance, its inhibition in combination with other treatments constitutes a valuable strategy to overcome this resistance [119]. CK2 inhibition may also be beneficial in an autoimmune disease setting, such as MS, as CK2 inhibition ameliorated EAE severity and incidence of relapses by the suppression of Th17 cells while promoting Tregs [114].

### 5.2. TGF-β

TGF-β is a pleiotropic cytokine that induces immune tolerance by regulating multiple types of immune cells [120,121]. It is expressed in various cell types, including immune cells and nonhematopoietic cells [122]. TGF-β is dysregulated in cancer patients and negatively regulates T and NK cell activity. It is secreted by glioblastoma cells and regulates cell proliferation, immunosuppression, angiogenesis, tumor invasion, and maintenance of the stemness of glioma stem cells through multiple signaling pathways [123]. Clinical study NCT00431561 showed that reducing TGF-β signaling by inhibiting mRNA translation through antisense oligonucleotides improves disease prognosis when combined with chemotherapy [123,124,125]. This was manifested by complete or partial remission of the tumor after almost one year, with a robust lesion size reduction. Another multicenter phase Ib/IIa clinical trial (NCT01220271) with galunisertib, a small molecule inhibitor of TGF-β kinase receptor type I, also showed an improvement in median overall and progression-free survival of glioma patients when administered in parallel to radio- and chemotherapy (with TMZ) [126]. However, the improvements were minimal, indicating alternative compensatory pathways mediated by other activators of downstream signaling [127].

For a synergistic response with improved efficiency, clinicians should consider a simultaneous approach including other targets, such as aberrant immune checkpoints. It is crucial to maintain a certain level of TGF-β, because the absence of this cytokine leads to the development of autoimmune diseases; in a murine T1D model, excessive Th1 responses and dysregulated Treg cell homeostasis occurred [128]. Previously, a phase I clinical trial with progressive MS patients showed that systemic application of recombinant TGF-β2 causes reversible nephrotoxicity, but did not improve disease outcome [129]. In Crohn’s disease patients, TGF-β signaling was restored in the intestine when Smad7 antisense oligonucleotides were taken to degrade Smad7 mRNA (Mongersen, GED-0301) [130]. Taking into consideration the complexity of these diseases, the abundant expression of TGF-β in the gut, which has been shown to directly affect the immune system via the gut–brain–immune axis, and the involvement of this molecule in CNS inflammation and repair, TGF-β holds potential as a therapeutic target [131,132].

### 5.3. TRAIL (CD253)

TRAIL/Apo2L, one of the two major cytokines of the TNF superfamily, acts through its TRAIL receptor subtypes: death receptors (DR) TRAIL-R1/DR4 and TRAIL-R2/DR5, and decoy receptors TRAIL-R3 and TRAIL-R4 [29,133]. This ligand is expressed at the surface of the two main immune effector cells, namely activated T cells and NK cells, but also on macrophages, neutrophils, and DCs [134]. TRAIL induces apoptosis by binding to its cognate receptors and the subsequent recruitment of adaptor proteins, which eventually initiate caspase-mediated signaling that leads to programmed cell death. TRAIL-mediated T cell cytotoxicity supports the elimination of tissue that is recognized as non-self, e.g., cancer cells or virus-infected tissue [29].

Cytokines communicate with the immune system and allow for an intercellular communication among tumor and parenchymal cells [135]. In glioma, TRAIL acts by the selective induction of cell death in malignant cells, while other cells are spared [136]. Under physiological conditions, TRAIL is not expressed in adult human brain tissue, but the apoptosis-mediating and truncated TRAIL receptors have been detected [137]. Under pathological conditions, activated CD4-positive cells employ TRAIL to selectively kill glioma cells [138]. Consequently, clinical trials targeting TRAIL have been conducted in glioma patients. In cancer patients, phase I–III clinical trials using agonistic mAbs that engage the TRAIL receptors DR4 and DR5 yielded promising results. Several patients with refractory or heavily pretreated disease have experienced stable disease upon treatment with anti-DR5 mAb. Antibodies naturally activate immune responses via Fc receptors [139]. Poly (ADP-ribose) polymerase (PARP) is involved in DNA repair and is responsible for DR-mediated extrinsic apoptotic signaling pathways [140]. Olaparib is a potent PARP inhibitor that overcomes apoptotic resistance and sensitizes glioblastoma cells for DR-mediated apoptosis induced by TRAIL. Currently, a phase I/IIa study of combined radiotherapy with olaparib and TMZ in high-grade glioma patients is underway (NCT03212742) [141].

TRAIL and TRAIL receptor knock-out mice display an increased disease severity in different models of induced autoimmune diseases, suggesting a protective role in autoimmunity [142,143,144,145]. However, the role of TRAIL remains controversial, as it has been shown that TRAIL blockade within the CNS suppresses MS in an EAE model by the inhibition of brain cell apoptosis [146]. Consistently, TRAIL-expressing T cells are not susceptible to cell death induced by this molecule [147]. This evidence of a dual role for TRAIL in the EAE model suggests that the selective blockade of TRAIL within the CNS and enhanced TRAIL function outside of the CNS may be required for its therapeutic value in MS patients. Challenges remain as TRAIL is involved in the death of primary cells, such as immune cells or neurons. Therefore, the mode of administration and the molecule design need to be approached cautiously in the treatment of glioblastoma and autoimmune diseases.

### 5.4. VEGF

VEGF is a proangiogenic agent produced mainly by endothelial cells, fibroblasts, smooth muscle cells, and macrophages [148], but VEGF has been shown to be upregulated by glioma cells as well. In fact, 80% of primary gliomas express VEGF-A and are, therefore, susceptible to anti-VEGF therapy [149]. Furthermore, immune modulation in the tumor microenvironment by antiangiogenic agents has been suggested by preclinical data and prompted clinical trials aiming at the dual blockade of VEGF and immune checkpoint molecules in different tumors [150]. Malignant brain tumors disrupt the physiological brain vasculature and, therefore, anti-VEGF treatment of glioma is still regarded as a promising therapy. Such treatment manifested in pruned and normalized tumor vasculature, alleviation of brain edema, and improved outcome of first-line therapies, such as radiation [151]. Furthermore, gliomas are characterized by immune evasion alongside excessive angiogenesis [151,152]. Currently, several trials administering anti-VEGF alone or in combination with other treatments, such as radiation or application of EGFR inhibitor (NCT01743950, NCT01884740, and more), are pending completion. In addition, clinical trials targeting VEGF receptors (VEGFRs) by peptide vaccination in combination with chemo- and radiotherapy were approved and yielded synergistic effects of these treatment options (UMIN000013381). Preliminary results revealed the safety and immunogenicity of this treatment. Additionally, two out of four patients showed complete remission upon treatment. As a result of the vaccination, CTLs became activated and attacked tumor blood vessels and cells [153]. Moreover, anti-VEGF plus anti-PD1 antibodies have been combined with chemo- and radiotherapy in a case report of a patient with recurrent glioma. Treatment proved to be safe and efficacious; however, a follow-up trial seems required [154]. Antiangiogenic-targeting therapies have achieved striking improvements in radiographic response, with high remission and survival rates. Hence, the optimization of such approaches to treat patients with recurrent glioblastoma is highly encouraging.

Antiangiogenic therapies may also be valuable for supporting treatment of autoimmune diseases. Clinical trials targeting VEGF in autoimmune diseases are already ongoing, e.g., the VEGF inhibitor bevacizumab (Avastin^®^) has been administered as an add-on therapy to high doses of corticosteroids for the treatment of acute optic neuritis and/or transverse myelitis in neuromyelitis optica (NMO) and neuromyelitis optica spectrum disorder (NMOSD) (NCT01777412). This combinatorial regimen proved to be beneficial for NMO/NMOSD patients presenting with an acute relapse [155]. Furthermore, trial NCT04311606 studies the beneficial effects of anti-VEGF treatment for patients with acute thyroid eye disease.

### 5.5. CD20

CD20 is a surface molecule found on most healthy and malignant B cells. The natural ligand of CD20 continues to elude detection. However, CD20 is associated with the BCR complex that suggests a role in BCR signaling, either by acting as a calcium channel or by directly modulating the BCR [30]. CD20 is a valuable target for mAbs, because the absence of a natural ligand means no known endogenous binding competitors. It also maintains stable binding epitopes by undergoing minimal post-translational modification [156].

There were several mAbs approved in the last decades for a variety of B cell malignancies, including rituximab, obinutuzumab, ofatumumab, and ocrelizumab. Rituximab has been administered to patients with CNS lymphoma alongside TMZ. The respective study claims that rituximab may sensitize B-lymphoma cells to the cytotoxic effects of TMZ [157]. Experimental studies with rituximab revealed that it might be involved in complement-dependent cytotoxicity, antibody-dependent cell-mediated cytotoxicity, and inhibition of cell proliferation [158]. Almost half of glioma patients show a B cell tumor infiltration that is distinguished by (i) immunosuppressive activity towards cytotoxic T cells, (ii) overexpression of inhibitory molecules PD-L1 and CD155, and (iii) production of immunosuppressive cytokines, such as TGF-β and IL-10 [159,160]. Application of an anti-CD20 immunotherapy provided an extended animal survival in glioma-bearing mice [159]. Therefore, a B-cell-depleting immunotherapy, such as rituximab, might prove beneficial in the GBM microenvironment that is overtaken by B-cell–mediated immunosuppression. Yet, the role of tumor infiltrating B cells in glioma must be further elucidated. Development of an autoimmune disease due to use of CD-20 mAbs is not to be expected. The mAb rituximab exerts its effect by depleting mainly CD20-expressing B cells from the circulation, thereby indirectly suppressing T cell activity [161]. It constitutes a second-line immunotherapy in multiple autoimmune-initiated disorders, and is often used as a therapy in patients with immune-mediated neurological disorders, where it shows long-term safety and efficacy. These include relapsing–remitting MS, autoimmune neuropathies, NMO, or myasthenia gravis [162,163,164]. The clinical success of rituximab and the aforementioned immune checkpoint regulators hold immense therapeutic potential (as summarized in Table 1).

## 6. Combination Therapies

Innovative combinations of drug regimens are being actively pursued in glioma therapy, as they have proven to be more effective than individual treatments. The classical radio- and chemotherapy approaches that often end in therapeutic resistance and insufficient targeting of glioma stem cells require synergistic regimens to improve their efficacy [165]. Radiotherapy directly induces cell death while enhancing immunogenicity; it contributes to BBB damage and leads to phenotypic changes in glioma cells. When combined with immune check point modulators, it elicits immunological effects without hindering their potency [166]. It is one of the most interesting combinatory strategies to address the poor GBM survival time [167]. Beside radiotherapy, antigen priming by vaccination shows beneficial improvements by enhancing the efficacy of antigen presentation in rescued T cells, which synergistically augments both antigen recognition and effector function. This approach is especially important to tackle one of the main challenges in GBM therapy, namely, low mutational burden and reduced availability of cognate antigens [166]. Vaccination with heat shock protein peptide complex 96 (HSPPC-96) showed, in a clinical trial (phase II), that the delivery of various tumor antigens causes an antitumor inflammatory response, which is safe and efficacious in recurrent GBM [168]. Additionally, novel immunotherapies should also address mechanisms of immune escape, including T cell exhaustion and adaptive resistance. This may be achieved by adding CAR T cells and CAR NK cells to the therapy plan. Several clinical trials using CAR-T-cell therapy against the GBM surface antigens IL13Ra2 and EGFRVIII appears safe and feasible [169,170]. Interestingly, these glioma antigens are considered nonimmune specific biomarkers [171]. Adoptive lymphocyte transfer (ALT) is another antigen-specific approach, whereby TILs are obtained from tumor specimens, altered by genetic engineering in vitro, and then sent back into the tumor site [172,173]. Clinical trials using oncolytic viruses that selectively infect tumor cells and induce tumor lysis revealed an improved survival rate of glioma patients [174]. However, valid viral spread and replication potency can be resisted by cancer stem cells and innate immune cells within the GBM microenvironment [175]. Another lymphocyte-targeted treatment includes bispecific T cell engagers (BiTEs). It consists of two single-chain variable fragments of different antibodies: one that binds to T cells via CD3, and the other to specific antigens expressed on the surface of tumor cells. The advantage of BiTE therapy is that it is produced and used without patient-specific individualization. Moreover, it has already been approved by the FDA to treat liquid malignancies and, recently, for GBM patients (NCT04903795) following promising preclinical results [176,177].

Ultimately, it is necessary to dissect the signaling networks and molecular players in GBM to better develop successful combination regimens and to assess the potential side effects that may result from drug combinations. The successful translation of drug combinations in the clinic is essentially due to the usage of lower doses of individual drugs, since combination therapy works synergistically or in an additive manner, thereby reducing the problems of drug resistance from the tumor or toxicity to healthy cells.

## 7. Conclusions

The immune response is guided by a series of checks, and costimulatory and coinhibitory pathways that—if imbalanced—may lead to a breakdown of self-tolerance and, thus, to autoimmunity. When the magnitude of the immune response exceeds the norm, a two-way road is possible, triggering either autoimmune disorders or cancer. The current approach in cancer therapy is to eliminate the block of the immune system to create autoimmune-like conditions. As such, their comorbid presentation creates a paradox regarding how such malignancies must be tackled therapeutically [178].

Several challenges arise in the treatment of glioma with immune checkpoint modulators, owing to its dynamic and immunosuppressive tumor microenvironment, the intra- and intertumor heterogeneity between patients, and the immunoselective blood–brain barrier impairing the ability of peripheral lymphocytes to traffic to the tumor mass. Immune-related adverse effects can produce life-threatening organ-specific damage, such as hepatotoxicity, cardiotoxicity, neurotoxicity, and thyroid insufficiency [9,35,179,180], or manifest in generalized symptoms, like fatigue or fever [180,181]. Nevertheless, most adverse reactions are manageable by discontinuation of the treatment and the administration of steroids or other biological antibodies [180,182].

The currently available tools make it difficult to predict immune-therapy-related adverse events from chemotherapy-related toxicities. Consequently, preventive surveillance strategies must be adapted. Risk factors for immune-related adverse events have been suggested, such as body mass index, gender, or the baseline neutrophil–lymphocyte ratio [183]. The dynamics of other biological biomarkers, like asymptomatic increases in creatinine kinase, elevations in liver enzymes, inflammatory cytokines, and autoantibodies, must be monitored in each therapy regime [182]. Previous findings suggest that preexisting T cell exhaustion may be a negative predictive biomarker of response to checkpoint inhibition [184]. Biomarkers that indicate inflammation can be either nonspecific or specific (organ- or drug-specific). C-reactive protein produced by the liver is one nonspecific biomarker that is generally an indicator of systemic inflammation [181]. For glioma, biomarkers such as cytokine, tumor cell surface antigens, or genetics (e.g., isocitrate dehydrogenase) should be used to evaluate progress and severity of the immune checkpoint therapies [171]. Identification of additional new prognostic and predictive biomarkers is crucial to enhance the outcome of immunotherapies. 

Using drug combinations rather than mono-immunotherapies, such as antagonistic mAbs against different immune receptors to achieve better clinical outcomes in patients, is an advance that has been successfully translated into therapy in glioma. However, the risk of developing or aggravating existing autoimmune diseases remains. A more promising option would be the additional combination with other first-line modalities, including chemotherapy, radiotherapy, vaccination, or oncolytic viruses. Combination therapy involving the above-mentioned immunotherapies and immune checkpoint inhibitors elicits immunotherapeutic benefits, while impeding the impact of tumor heterogeneity and T cell exhaustion [175,185]. Synergistically targeting cancer therapy must focus on the heterogeneous tumor and its dynamic microenvironment, including specific biomarkers while not neglecting the optimal tactics to prevent autoimmunity.

## Figures and Tables

**Figure 1 cancers-13-03524-f001:**
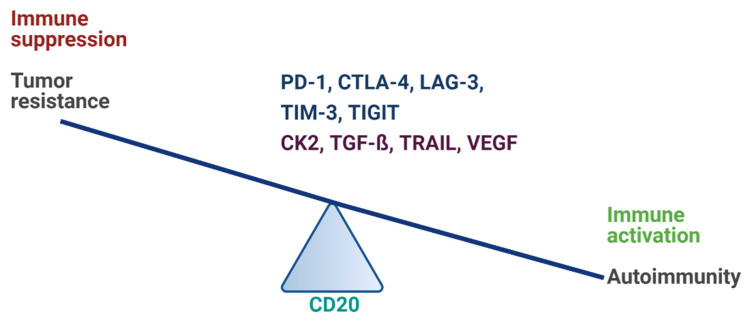
Classical and nonclassical immune modulators in malignant glioma and autoimmune diseases. A careful balance should be maintained when targeting immune modulators in therapeutic efforts, whereas B-cell-depleting immunotherapy (by CD20 blockade) may be beneficial in both malignant glioma and autoimmune diseases. Classical molecules (blue)—PD-1: programmed cell death protein-1, CTLA-4: cytotoxic T-lymphocyte antigen 4, LAG-3: lymphocyte-activation gene 3, TIM-3: T-cell immunoglobulin and mucin-domain containing-3, TIGIT: T cell immunoreceptor with Ig and ITIM domains. Nonclassical molecules (purple)—CK2: casein kinase 2, TGF-β: transforming growth factor beta, TRAIL: tumor necrosis factor-related apoptosis-inducing ligand, VEGF: vascular endothelial growth factor, CD20: cluster of differentiation 20 (B-lymphocyte antigen). Figure created with BioRender.com (accessed on 28 May 2021).

**Figure 2 cancers-13-03524-f002:**
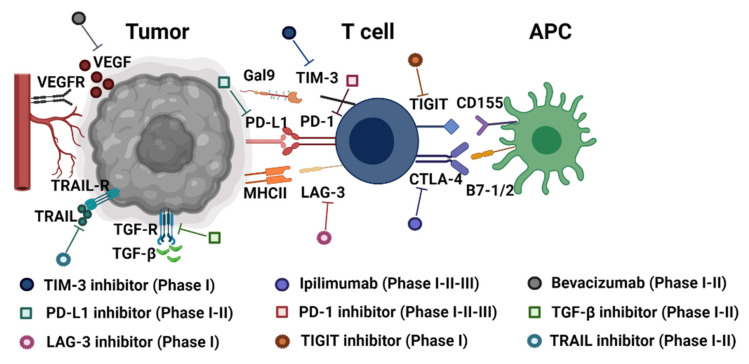
Targeting immune checkpoint molecules in malignant glioma. Immune pathways and actions of checkpoint inhibitors that are necessary to maintain antitumor activity. Immune checkpoint molecules are expressed on the surface of T cells and interact with their ligands on antigen-presenting cells (APCs; such as dendritic cells). The phase of clinical trials targeting each immune checkpoint molecule in glioma patients is indicated as I, II, or III. VEGFR: vascular endothelial growth factor receptor, VEGF: vascular endothelial growth factor, TRAIL-R: TNF-related apoptosis-inducing ligand receptor, TRAIL: TNF-related apoptosis-inducing ligand, TGF-R: transforming growth factor beta receptor, TGF-β: transforming growth factor beta, Gal-9: galectin 9, TIM-3: T-cell immunoglobulin and mucin-domain containing-3, PD-L1: programmed cell death ligand-1, PD-1: programmed cell death protein-1, MHCII: major histocompatibility complex II, LAG-3: lymphocyte-activation gene 3, CTLA-4: cytotoxic T-lymphocyte antigen 4, B7: co-stimulation ligand, TIGIT: T cell immunoreceptor with Ig and ITIM domains, CD155: cluster of differentiation 155. Figure created with Biorender.com (accessed on 2 July 2021).

**Table 1 cancers-13-03524-t001:** Clinical trials of classical and nonclassical immune modulators in autoimmune diseases and glioma.

Targeted Molecule	G	AD	Phase	Treatment	Study Number
PD-1	X		III	E: Nivolumab + TMZ + RT C: Nivolumab Placebo + TMZ + RT	NCT02667587
	X		II	Neoadjuvant Nivolumab	NCT02550249
	X		II	Prior in all groups: RT + TMZ E: Nivolumab + TMZ Control: TMZ alone	NCT04195139
CTLA-4	X		II/III	E: Nivolumab + Ipilimumab + RT C: TMZ + RT	NCT04396860
		X	II	E: Abatacept followed by placebo C: placebo followed by Abatacept	NCT01116427
	X		II	E: Ipilimumab + Nivolumab followed by Nivolumab alone	NCT04145115
		X	I/II	E1: CTLA4-Ig + Cyclophosphamide E2: CTLA4-Ig + Cyclophosphamide Control: Cyclophosphamide alone	NCT00094380
	X		I	E1: Nivolumab + placebo followed by Nivolumab alone E2: Nivolumab + Ipilimumab followed by Nivolumab alone E3: Placebo + Ipilimumab followed by Nivolumab alone	NCT04323046
		X	I	E: CTLA4-Ig	NCT00076934
	X		I	E1: Nivolumab + Ipilimumab followed by IL13Ralpha2-CAR T cells + Nivolumab E2: IL13Ralpha2-CAR T cells + Nivolumab	NCT04003649
		X	I/II	E: Belatacept/Abatacept (multiple doses)	NCT00279760
	X		I	E: Ipilimumab (intra-tumoral) + Nivolumab (intravenous)	NCT03233152
LAG-3	X		I	E A1: anti-LAG-3 E A2: Urelumab E B1: anti-LAG-3 + Nivolumab E B2: Nivolumab + Urelumab E I: patients receive pre-operatively and 45 days after surgery a drug from one of the four arms mentioned above	NCT02658981
	X		I	E: anti-LAG-3 + Nivolumab	NCT03493932
		X	Prep	anti-LAG-3 (patent no. 3344654)	
CK2	X		I/II	CK-2 inhibitor in recurrent medulloblastoma E I: children E II: adults E S: before surgery in subjects from I and II	NCT03904862
TIGIT	X		0/I	E A: anti-TIGIT + anti-PD-1 (Safety Cohort) E B1: anti-TIGIT + placebo (Surgical Cohort) E B2: anti-PD-1 + placebo (Surgical Cohort) E B3: anti-TIGIT + anti-PD-1 (Surgical Cohort) E B4: placebo (Surgical Cohort) all Experimental B followed by anti-TIGIT + anti-PD-1	NCT04656535
TIM-3	X		I	E: anti-TIM-3 + anti-PD-1 + radiation therapy	NCT03961971
TGF-β	X		Ib/IIa	E IA: RT + 80 mg TGF-β inhibitor + TMZ followed by TGF-β inhibitor + TMZ E IB: RT + 150 mg TGF-β inhibitor + TMZ followed by TGF-β inhibitor + TMZ E II: RT + established dose from I of TGF-β inhibitor + TMZ followed by TGF-β inhibitor + TMZ Control: RT + TMZ followed by TMZ alone	NCT01220271
TRAIL	X		I/IIa	E: Olaparib + TMZ + RT followed by Olaparib alone, then Olaparib + TMZ	NCT03212742
VEGF		X	II	E 1: saline + Aflibercept E 2: hyaluronidase + Aflibercept E 3: hyaluronidase alone	NCT04311606
	X		II	C 1: Bevacizumab + radiation (naive recurrent grade IV gliomas) C 2: Bevacizumab + radiation (exposed and refractive grade IV gliomas) C 3: Bevacizumab + radiation (naive recurrent grade III gliomas) C 4: Bevacizumab + radiation (exposed and refractive grade III gliomas)	NCT01743950
	X		I/II	E: Cetuximab + Bevacizumab via Superselective Intraarterial Cerebral Infusion	NCT01884740
	X		I/II	E: peptide vaccine of VEGFR (subcutaneous)	UMIN000013381
CD20		X	IV	E: Rituximab (standard infusion + rapid infusion)	NCT02040116
		X	III	C 1: Rituximab (infusion) C 2: Cladribine (oral)	NCT04121403
		X	II	E: Rituximab followed by Rituximab C: Placebo followed by Rituximab	NCT04274257
		X	II	E 1: Rituximab (intrathecal) + methylprednisolone (intravenous) E 2: Rituximab (intrathecal) + methylprednisolone (intravenous) + Rituximab (intravenous) C: methylprednisolone (intravenous)	NCT02545959
		X	II	E: Rituximab (intravenous) C: placebo (intravenous)	NCT00279305
		X	I/II	E: Rituximab (intravenous)	NCT00036491
		X	I	E: Rituximab (2 times intravenous)	NCT01086631
		X	I	E: Rituximab (2 times intravenous)	NCT00101829

G: glioma, AD: autoimmune disease, E: experimental, C: comparator, TMZ: temozolomide, RT: radiotherapy, Prep: in preparation.

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
