# Peer review of "Targeting Immune Modulators in Glioma While Avoiding Autoimmune Conditions"

_cancers, 2021, doi:10.3390/cancers13143524_

Round 1

Reviewer 1 Report

The manuscript describes the importance of immune check point modulators in the treatment of glioma. An additional aim of the review is to focus the attention on autoimmune diseases.

Authors should better describe in the introduction what is the association between glioma, immune check point modulators and autoimmune diseases. Is there a link between glioma and autoimmune diseases? is the usage of immune check point modulators responsible for the development of autoimmune diseases in glioma patients??

Some images about the pathways induced by immune check points should be added

References should be revised. Some recent papers about glioma and immune check points are not cited (De Felice, 2019; Kamran, 2016)

Reviewer 2 Report

GBM is the most common primary malignant brain tumour. Survival is quite poor and more efficacious treatment options are urgently needed. Although immunotherapies have emerged as effective treatments for a number of cancers, translation of these through to brain tumours is a distinct challenge.

In the review, the authors discussed encouraging immune-modulatory targets in clinical research in glioblastoma.

The review presents interesting and valuable aspects of glioblastoma therapy. The review is well written and elaborated. However, few points require further attention:

  1. The authors focused on describing checkpoint inhibitors and some modulators in glioblastoma therapies. However, what would be important, is to further elaborate: combinatory therapies, CAR, BiTE, TILs, oncolytic virotherapy, since novel therapies becoming more and more advanced and promising in solid cancer therapies.
  2. Having a table with a general overview of various treatment modalities, including SoC/treatment scheme would be informative for potential readers.
  3. What are the current biggest challenges and limitations in glioblastoma therapies? 
  4. Introduction to the biomarkers and their potential role in treatment discoveries, predictive or prognostic value in GBM therapies would increase the quality of the paper.
  5. Conclusions focusing in more detail on future prospects should be elaborated.

Round 2

Reviewer 1 Report

The revised version of the manuscript is suitable for publication.